# A Robust Method for the Estimation of Kinetic Parameters for Systems Including Slow and Rapid Reactions—From Differential-Algebraic Model to Differential Model

**Tapio Salmi [1],*, Esko Tirronen [1], Johan Wärnå [1], Jyri-Pekka Mikkola [1,2], Dmitry Murzin [1] and Valerie Eta [1]**

[1]   Labradory of Industrial Chemistry and Reaction Engineering, Johan Gadolin Process Chemistry Centre, Åbo Akademi University, FI-20500 Turku, Finland; esko.tirrFonen@gmail.com (E.T.); johan.warna@abo.fi (J.W.); jpmikkol@abo.fi (J.-P.M.); dmitry.murzin@abo.fi (D.M.); valerie.eta@rebio.fi (V.E.)

[2]   Department of Chemistry, Technical Chemistry, Chemical-Biological Center, Umeå University, SE-90187 Umeå, Sweden

*   Correspondence: tapio.salmi@abo.fi

**Abstract:** Reliable estimation of kinetic parameters in chemical systems comprising both slow and rapid reaction steps and rapidly reacting intermediate species is a difficult differential-algebraic problem. Consequently, any conventional approach easily leads to serious convergence and stability problems during the parameter estimation. A robust method is proposed to surmount this dilemma: the system of ordinary differential equations and nonlinear algebraic equations is converted to ordinary differential equations, which are solved in-situ during the parameter estimation. The approach was illustrated with two generic examples and an example from green chemistry: synthesis of dimethyl carbonate from carbon dioxide and methanol.

**Keywords:** kinetics; slow and rapid reactions; robust parameter estimation; dimethyl carbonate

## 1. Introduction

A reliable estimation of kinetic and thermodynamic parameters is one of the most important tasks in chemical reaction engineering. Only in few cases, like in the case of some homogeneous gas-phase reactions, is it possible to determine the kinetic constants a priori, exclusively from theoretical calculations. Consequently, in most cases, an extensive matrix of experimental work is needed. Particularly, in the presence of heterogeneous catalysts, small variations in the chemical composition and physical structure of the catalyst can change the reaction kinetics, and a new experimental program is inevitable. Typically, kinetic experiments are carried out in batch reactors or in continuous reactors with a well-established flow pattern, i.e., perfect back-mixing or plug flow. After that, the kinetic and thermodynamic parameters are determined by non-linear regression analysis. In the most recent 30 years, numerical methods and computing power have advanced tremendously. The development of solvers for stiff differential equations has enabled the treatment of difficult problems in chemical kinetics and simulation of large chemical systems appearing in combustion and atmospheric processes. However, even a superficially small problem can become a difficult one, as the system usually consists of a set of slow and rapid reaction steps. From a mathematical viewpoint, the system of ordinary differential equations describing the behavior of the components in a batch reactor becomes a system of differential-algebraic equations (DAE), for which solvers have been developed in recent years. In principle, the task could be solved by coupling a DAE solver to a parameter estimator

(a nonlinear regression routine). However, if the parameters glide to an unrealistic regime in the course of the parameter estimation, significant problems arise: the solutions of the algebraic equations induce problems, such as negative concentrations, which usually leads to the collapse of the whole computational process. Our experience with different case studies [1] has taught that a more robust approach is to convert the DAE problem to a set of ordinary differential equations (ODEs) and then solve these ODEs as a sub-problem in the parameter estimation. Here we illustrate the method with two generic examples and a highly relevant issue in chemical technology, namely the utilization of carbon dioxide as a raw material for the synthesis of green chemicals.

## 2. Development of a Robust Method

The tasks in the method are briefly summarized as follows:

T1. Mass balances of all components in the batch reactor are written down (ODEs)
T2. A quasi-steady-state hypothesis is applied to the intermediates—the ODEs become a system of DAEs is created
T3. The DAE system is converted to a set of implicit ODEs by differentiation
T4. The system of implicit ODEs is converted to a set of explicit ODEs
T5. The system of explicit ODEs is implemented in a combined stiff ODE solver—parameter estimation software Modest.

### 2.1. Example: Consecutive Reactions with Slow and Rapid Steps

The following consecutive reaction scheme is considered in Figure 1. Step 1 is presumed to be slow whereas step 2 is rapid.

$$A \underset{}{\overset{1}{\rightleftharpoons}} R \underset{}{\overset{2}{\rightleftharpoons}} S$$

**Figure 1.** Consecutive reaction system.

The mass balances and rate equations of this consecutive reaction system can be written in the general case as follows,

$$\frac{dC_A}{dt} = -r_1 \tag{1}$$

$$\frac{dC_R}{dt} = r_1 - r_2 \tag{2}$$

$$\frac{dC_S}{dt} = r_2 \tag{3}$$

$$r_1 = k_1\left(C_A - \frac{C_R}{K_1}\right) \tag{4}$$

$$r_2 = k_2\left(C_R - \frac{C_S}{K_2}\right) \tag{5}$$

For the special case, where reaction step 1 is slow and step 2 is rapid, a reduced model can be derived:

$$\frac{C_S}{C_R} = K_2 \tag{6}$$

$$\frac{dC_S}{dt} = K_2\frac{dC_R}{dt} \tag{7}$$

$$\frac{dC_A}{dt} + \frac{dC_R}{dt} + \frac{dC_S}{dt} = 0 \tag{8}$$

Substituting $\frac{dC_S}{dt}$ into Equation (8) yields

$$\frac{dC_A}{dt} + (1 + K_2)\frac{dC_R}{dt} = 0 \tag{9}$$

$$\frac{dC_R}{dt} = -\frac{1}{1 + K_2}\frac{dC_A}{dt} = +\frac{1}{1 + K_2}\, r_1 \tag{10}$$

$$\frac{dC_S}{dt} = +\frac{K_2}{1 + K_2}\, r_1 \tag{11}$$

As a summary we obtain for the special case of the consecutive reaction system, where step 1 is slow and step 2 is rapid:

$$\frac{dC_A}{dt} = -r_1$$

$$\frac{dC_A}{dt} = -r_1$$

$$\frac{dC_R}{dt} = \frac{1}{1 + K_2}\, r_1$$

$$\frac{dC_S}{dt} = \frac{K_2}{1 + K_2}\, r_1$$

$$r_1 = k_1\left(C_A - \frac{C_R}{K_1}\right)$$

The general case was compared with the special case by increasing the value of the kinetic constant $k_2$ of the rapid reaction step 2 to find the value of $k_2$ by which the solution of the general case approaches that of the special case.

By setting $k_1 = 1$ and assuming arbitrary values for the equilibrium constants $K_1 = 10$ and $K_2 = 10$, the concentration profiles depicted in Figure 2 were obtained for the special case.

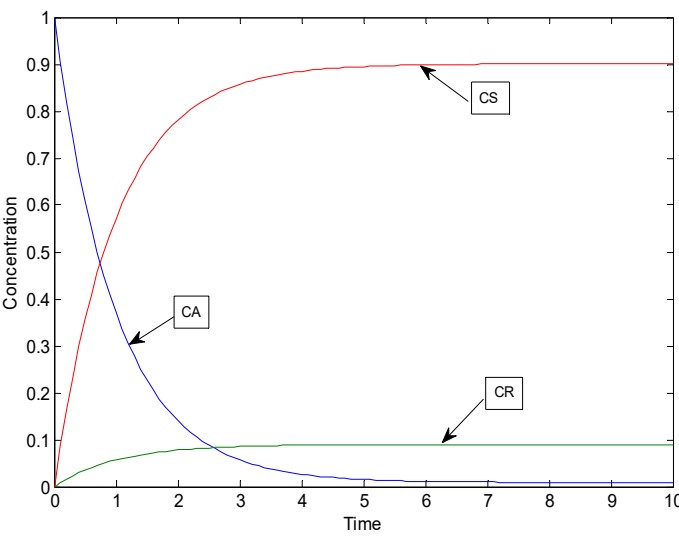

**Figure 2.** Concentration profiles of the special case of consecutive reactions with parameters: $k_1 = 1$, $K_1 = 10$ and $K_2 = 10$.

As revealed by Figure 2, the system behaves like a parallel reaction system, which is caused by the fact that as soon as R is formed, some part of it is immediately transformed to S.

The general case was simulated by increasing the value of the rate parameter $k_2$ of the rapid reaction step 2. Other parameters $k_1 = 1$, $K_1 = 10$ and $K_2 = 10$ were kept the same as in the special case. The sum of squared residuals was calculated for the difference between simulated data points of the

special case and the general case (number of simulated data points was 101 in the time frame of 0–10 in each case). Concentration profiles for the general case of consecutive reaction system corresponding to $k_2$ values of 1, 5, 10, 100 and 300 are depicted in Figure 3.

The sums of squared residuals of the general case simulations of consecutive reaction system with $k_1 = 1$, $k_2 = 1, 5, 10, 100$ and 300, $K_1 = 10$ and $K_2 = 10$ compared to the special case are presented in Table 1.

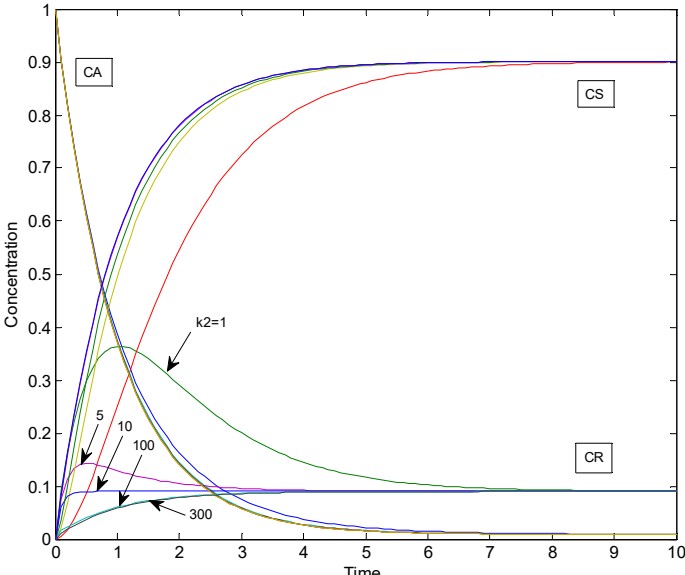

**Figure 3.** Concentration profiles of the general case of consecutive reactions with parameters: $k_1 = 1$, $k_2 = 1, 5, 10, 100$ and 300, $K_1 = 10$ and $K_2 = 10$.

The behavior of the sum of squared residuals as a function $k_2$ is displayed in Figure 4. The general case approaches the special case as the rate constant $k_2$ of the rapid reaction step 2 exceeds 100 i.e., 100 times the value of the rate parameter $k_1$ of the slow reaction step 1.

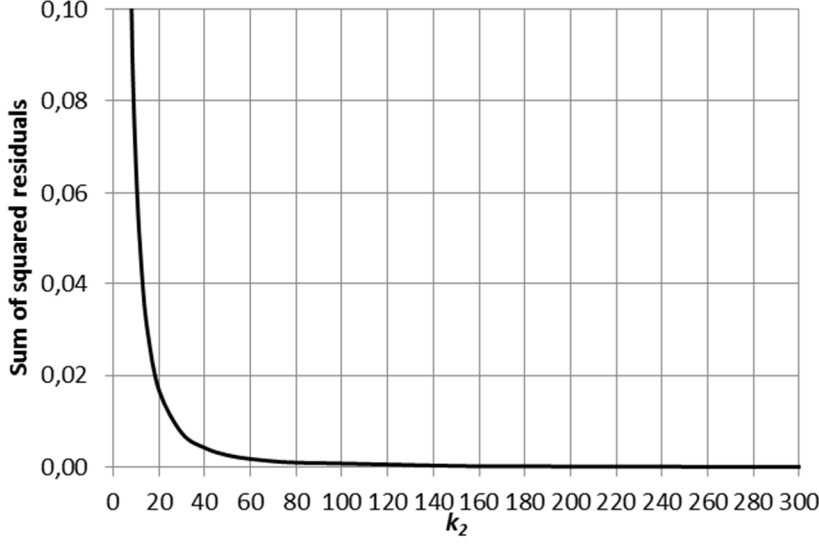

**Figure 4.** Sum of squared residuals as a function of the rate parameter $k_2$ of the rapid reaction step 2 in the general case of consecutive reaction system compared to the special case ($k_1 = 1$, $K_1 = 10$ and $K_2 = 10$).

**Table 1.** The sum of squared residuals of the general case simulations as compared to the special case ($k_1 = 1$, $K_1 = 10$ and $K_2 = 10$).

| $k_2$ | S |
|---|---|
| 1 | 3.6113 |
| 5 | 0.2373 |
| 10 | 0.0644 |
| 100 | 0.006522 |
| 300 | 0.00071645 |

*2.2. Parallel Reactions with Slow and Rapid Steps*

The second example is the classical parallel reaction scheme displayed in Figure 5. Step 1 is presumed to be slow while step 2 is rapid.

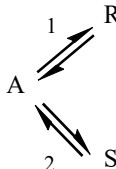

**Figure 5.** Parallel reaction system.

The mass balances and rate equations of this reaction system can be written in the general case as follows,

$$\frac{dC_A}{dt} = -r_1 - r_2 \tag{12}$$

$$\frac{dC_R}{dt} = r_1 \tag{13}$$

$$\frac{dC_S}{dt} = r_2 \tag{14}$$

$$r_1 = k_1\left(C_A - \frac{C_R}{K_1}\right) \tag{15}$$

$$r_2 = k_2\left(C_A - \frac{C_S}{K_2}\right) \tag{16}$$

For the special case, where reaction step 1 is slow and step 2 is rapid, a reduced model can be derived for the parallel reaction system:

$$\frac{dC_A}{dt} + \frac{dC_R}{dt} + \frac{dC_S}{dt} = 0 \tag{17}$$

$$\frac{C_S}{C_A} = K_2 \tag{18}$$

$$\frac{dC_S}{dt} = K_2\frac{dC_A}{dt} \tag{19}$$

Substituting $\frac{dC_S}{dt}$ into Equation (17) yields:

$$(1 + K_2)\frac{dC_A}{dt} + \frac{dC_R}{dt} = 0 \tag{20}$$

$$\frac{dC_A}{dt} = -\frac{1}{1 + K_2}\frac{dC_R}{dt} = -\frac{1}{1 + K_2}r_1 \tag{21}$$

$$\frac{dC_S}{dt} = -\frac{K_2}{1 + K_2} r_1 \tag{22}$$

At time $t = 0$,

$$C_{R(0)} = 0 \tag{23}$$

$$\frac{C_{S(0)}}{C_{A(0)} - C_{S(0)}} = K_2 \tag{24}$$

$$C_{S(0)} = \frac{K_2 C_{0A}}{1 + K_2} \tag{25}$$

$$C_{A(0)} = \frac{C_{0A}}{1 + K_2} \tag{26}$$

The initial conditions (25) and (26) arise from the fact that some amounts of component S is formed immediately in the system, because step 2 progresses with an infinite rate. As a summary we obtain for the special case of the parallel reaction system, where step 1 is slow and step 2 is rapid:

$$\frac{dC_A}{dt} = -\frac{1}{1 + K_2} r_1$$

$$\frac{dC_R}{dt} = r_1$$

$$\frac{dC_S}{dt} = -\frac{K_2}{1 + K_2} r_1$$

$$r_1 = k_1 \left( C_A - \frac{C_R}{K_1} \right)$$

$$C_{A(0)} = \frac{C_{0A}}{1 + K_2}$$

$$C_{R(0)} = 0$$

$$C_{S(0)} = \frac{K_2 C_{0A}}{1 + K_2}$$

Initial values $C_A(0)$ and $C_S(0)$, $C_R(0) = 0$ as a function of equilibrium constant $K_2$ in the special case of a parallel reaction system are depicted in Figure 6.

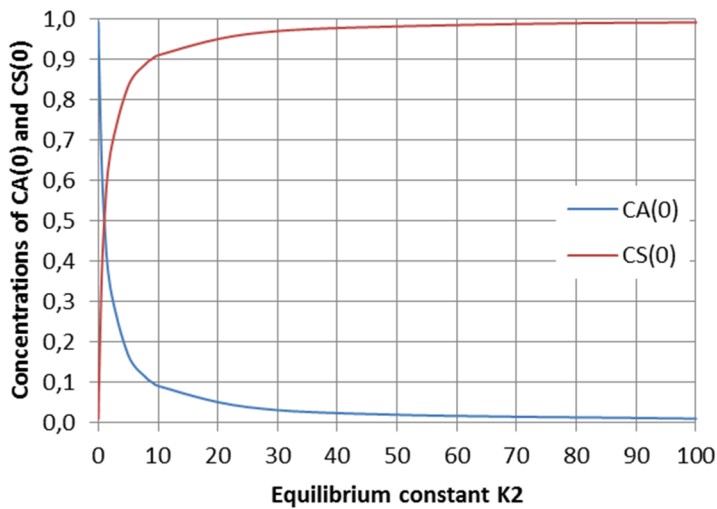

**Figure 6.** Initial values $C_A(0)$ and $C_S(0)$, $C_R(0) = 0$ as a function of equilibrium constant $K_2$ in the special case of a parallel reaction system, where step 1 is slow and step 2 is rapid.

Some concentration profiles as an example for the special case of a parallel reaction system with parameters $k_1 = 1$, $K_1 = 10$ and $K_2 = 2$ are displayed in Figure 7.

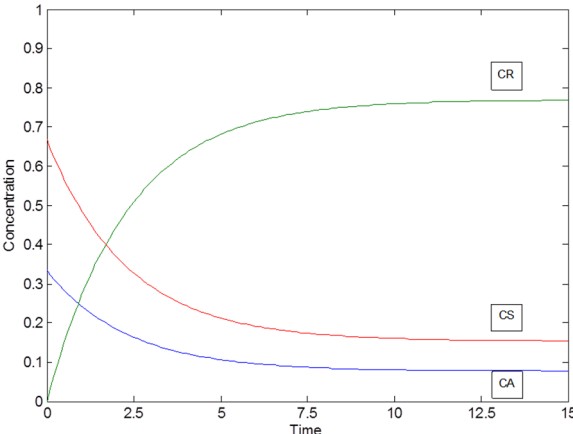

**Figure 7.** Concentration profiles of the special case of a parallel reaction system with parameters: $k_1 = 1$, $K_1 = 10$ and $K_2 = 2$.

The general case of a parallel reaction system was compared with the special case as in the previous example 2.1 by increasing the value of the rate parameter $k_2$ of the rapid reaction step 2 to find the value of $k_2$ by which the solution of the general case approaches that of the special case. Concentration profiles for the general case of parallel reactions corresponding to $k_2$ values of 2, 10 and 50 are depicted in Figure 8.

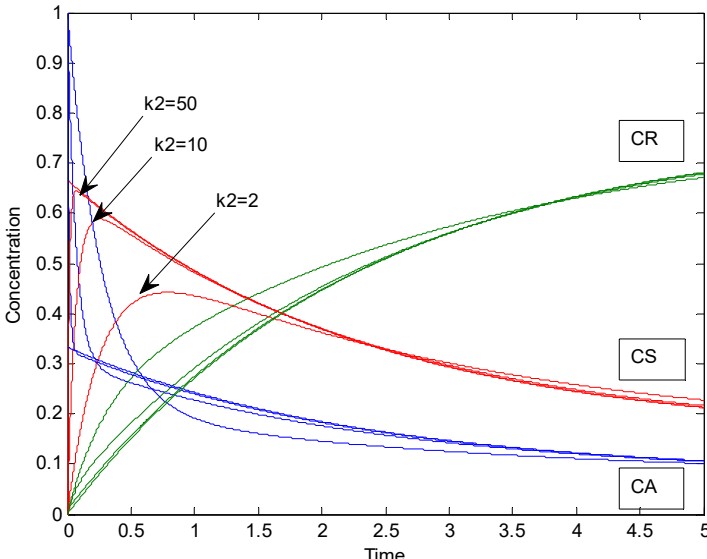

**Figure 8.** Concentration profiles of the general case of a parallel reaction system with parameters: $k_1 = 1$, $k_2 = 2$, 10 and 50, $K_1 = 10$ and $K_2 = 2$.

The sum of squared residuals calculated as in the example 2.1 between simulated data points of the special case and the general case with different values of the rate parameter $k_2$ while $k_1 = 1$, $K_1 = 10$ and $K_2 = 2$, (the number of simulated data points was 301 in the time interval of 0–15) are shown in Table 2. The behavior of the sum of squared residuals as a function is displayed in Figure 9. The general case approaches the special case in this example when $k_2$ is >50.

**Table 2.** The sum of squared residuals of the general case of parallel reaction system compared to the special case with different values of $k_2$ ($k_1 = 1$, $K_1 = 10$ and $K_2 = 2$).

| $k_2$ | S |
|---|---|
| 2 | 3.31901 |
| 5 | 1.68380 |
| 10 | 1.14954 |
| 50 | 0.89085 |
| 100 | 0.88929 |
| 1000 | 0.888893 |
| 10,000 | 0.888889 |

This kind of parallel reaction system has been studied by Branco et al. [2], who developed a concept for the determination of the switching point between kinetic and thermodynamic control for cases when one of the reactions is very rapid, whereas the second one is thermodynamically favored. The same phenomenon can be seen in Figure 6: S is the kinetically favored product, with a rapidly increasing concentration in the beginning, but with an increasing reaction time the concentration of R becomes higher because it is favored by thermodynamics, i.e., its formation has a higher equilibrium constant.

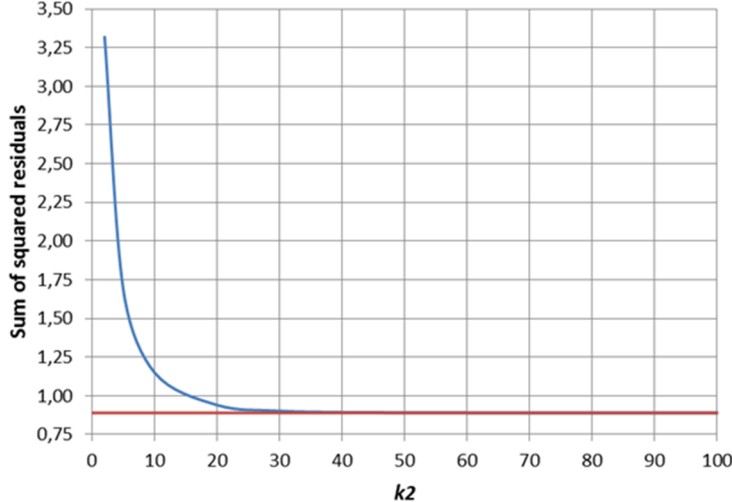

**Figure 9.** The sum of the squared residuals as a function of the rate parameter $k_2$ of the fast reaction step 2 in the general case of a parallel reaction system compared to the special case ($k_1 = 1$, $K_1 = 10$ and $K_2 = 2$).

### 2.3. Example: Synthesis of Dimethyl Carbonate from Methanol and Carbon Dioxide

Carbon dioxide reacts with methanol (MeOH) to yield dimethyl carbonate (DMC), which is a green alternative to methyl tert-butyl ether (MTBE) and can be used as a carbonylating and methylating agent [3,4]. The presence of a heterogeneous catalyst is required for the synthesis of DMC. We used zirconia-based catalysts ($ZrO_2$-MgO) in an isothermal and isobaric laboratory-scale batch reactor [5]. The thermodynamics for this reaction is extremely unfavorable, so a way to shift the equilibrium is to include an additive to the reaction mixture; the role of the additive was to act as a chemical dehydration agent, i.e., to capture the water formed in the reaction. Butylene oxide (BO) was selected as the additive [5]. In this way, the process gains a more irreversible character and can be forced to the side of the products. Methylene butylate (MB) appears as an intermediate species, forming butylene glycol (BG) and thus preventing the water formation. The reaction scheme is displayed below in Figure 10, where * denotes a vacant site on the surface of the catalyst, and MeOH* denotes adsorbed methanol on the catalyst surface.

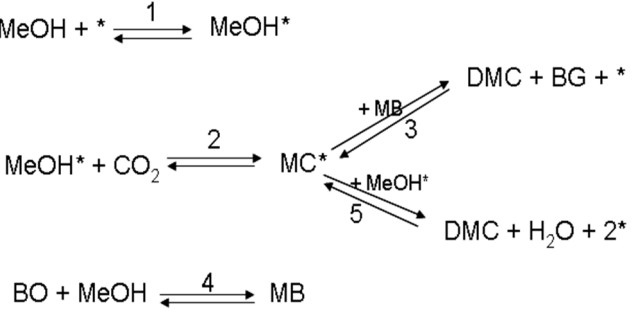

**Figure 10.** Reaction scheme for the synthesis of dimethyl carbonate (DMC).

Steps 1–3 and 5 take place on the catalyst surface, while step 4 proceeds as a homogeneous liquid-phase reaction. Reaction steps 2 and 4 in the scheme were assumed to be rate determining, whereas the adsorption step of methanol was presumed to be rapid. In addition, steps 3 and 5 were taken as rapid steps compared to steps 2 and 4. A constant-density system was assumed and the mass balance of carbon dioxide was not included, since $CO_2$ was constantly added to the system by keeping the pressure constant. Thus, the saturation concentration of $CO_2$ was presumed, and Henry's law was applied to relate the partial pressure of $CO_2$ and the concentration of dissolved $CO_2$. A large excess of methanol was used. Based on the reaction mechanism displayed above, the rate equations for the rate determining steps were derived. The details of the derivation of the rate equations are given as supplementary material in Appendix A: Derivation of the Rate Equations.

2.3.1. Basic Mass Balances

The mass balances for bulk phase components in a batch reactor (assuming constant density) at a constant $CO_2$ concentration due to controlled pressure can be written as follows ($\rho_B$ = mass of catalyst-to-liquid volume, i.e., the catalyst bulk density)

$$\frac{d[DMC]}{dt} = (r_3 + r_5)\rho_B \tag{27}$$

$$\frac{d[BG]}{dt} = r_3 \rho_B \tag{28}$$

$$\frac{d[H_2O]}{dt} = r_5 \rho_B \tag{29}$$

$$\frac{d[MeOH]}{dt} = -r_1 \rho_B - r_4 \tag{30}$$

$$\frac{d[BO]}{dt} = -r_4 \tag{31}$$

$$\frac{d[MB]}{dt} = -r_3 \rho_B + r_4 \tag{32}$$

Applying a quasi-steady state to MC* and MeOH* gives relations (33) and (34),

$$r_{MC*} = r_2 - r_3 - r_5 \approx 0 \tag{33}$$

$$r_{MeOH*} = r_1 - r_2 - r_5 \approx 0 \tag{34}$$

Substituting Equations (33) and (34) into Equations (27)–(32) results in the following relationships:

$$\frac{d[DMC]}{dt} = r_2 \rho_B \tag{35}$$

$$\frac{d[BG]}{dt} = r_3 \rho_B \tag{36}$$

$$\frac{d[H_2O]}{dt} = r_5 \rho_B \tag{37}$$

$$\frac{d[MeOH]}{dt} = -(r_2 + r_5)\rho_B - r_4 \tag{38}$$

$$\frac{d[BO]}{dt} = -r_4 \tag{39}$$

$$\frac{d[MB]}{dt} = -r_3 \rho_B + r_4 \tag{40}$$

### 2.3.2. Differential-Algebraic Problem

The addition of the mass balances further gives

$$\frac{d[MB]}{dt} + \frac{d[BG]}{dt} = r_4 \tag{41}$$

$$\frac{d[MB]}{dt} + \frac{d[MeOH]}{dt} = -2r_2 \rho_B \tag{42}$$

$$\frac{d[H_2O]}{dt} + \frac{d[BG]}{dt} = r_2 \rho_B \tag{43}$$

The ODEs (Equations (33) and (35)–(37)) give the stoichiometric relations

$$\int_0^{[DMC]} d[DMC] = \int_0^{[H_2O]} d[H_2O] + \int_0^{[BG]} d[BG] \tag{44}$$
$$[DMC] = [H_2O] + [BG]$$

Equations (36), (39) and (40) give

$$-\int_{[BO]_o}^{[BO]} d[BO] = \int_0^{[BG]} d[BG] + \int_0^{[MB]} d[MB] \tag{45}$$
$$[BO]_O = [BO] + [BG] + [MB]$$

From Equations (27), (30) and (32)–(34) is obtained

$$-2\int_0^{[DMC]} d[DMC] = \int_{[MeOH]_o}^{[MeOH]} d[MeOH] + \int_0^{[MB]} d[MB] \tag{46}$$
$$[MeOH]_O = [MeOH] + 2[DMC] + [MB]$$

The following relationship between the concentrations was obtained by consideration of the reaction mechanism [6] ($\alpha = K_1 K_5 / K_3$),

$$\frac{[DMC]}{[BG]} = 1 + \alpha \frac{[MeOH]}{[MB]} \tag{47}$$

Equations (35), (39), (41)–(43) and (47) form a set of six differential-algebraic equations, with six unknowns: ([DMC], [BO], [BG], [H$_2$O], [MB], [MeOH]).

From Equation (47), the following expression can be obtained:

$$\frac{[BG]}{[DMC]} = \frac{1}{1 + \alpha \frac{[MeOH]}{[MB]}} = \frac{[MB]}{[MB] + \alpha[MeOH]} \tag{48}$$

From Equations (44) and (47), the following relationship is obtained:

$$\frac{[H_2O]}{[DMC]} = 1 - \frac{[BG]}{[DMC]} \tag{49}$$

$$\frac{[H_2O]}{[DMC]} = \frac{\alpha[MeOH]}{[MB] + \alpha[MeOH]} \tag{50}$$

Equations (45) and (48) give the following relation:

$$\frac{[MB][DMC]}{[MB] + \alpha[MeOH]} + [MB] = [BO]_O - [BO] = \omega \tag{51}$$

i.e.,

$$[MB][DMC] + [MB]^2 + \alpha[MeOH][MB] = \omega[MB] + \alpha[MeOH]\omega \tag{52}$$

i.e.,

$$[MB]^2 + ([DMC] - \omega)[MB] + (\alpha[MB] - \alpha\omega)([MeOH]_O - 2[DMC] - [MB]) = 0 \tag{53}$$

Rearranging of Equation (53) gives

$$(1 - \alpha)[MB]^2 + ([DMC] - \omega + \alpha\gamma + \alpha\omega)[MB] - \alpha\gamma\omega = 0 \tag{54}$$

where $\gamma = [MeOH]_O - 2[DMC]$. The solution of the second-degree Equation (54) becomes

$$[MB] = \frac{\beta \pm \sqrt{\beta^2 + 4(1 - \alpha)\alpha\gamma\omega}}{2(1 - \alpha)} \tag{55}$$

where $\beta = \omega - [DMC] - \alpha(\gamma + \omega)$. The sign + in the nominator gives the physically meaningful solution of Equation (55).

Equations (46), (48) and (50) give

$$[MeOH] = \gamma - [MB] \tag{56}$$

$$[BG] = \frac{[MB][DMC]}{[MB] + \alpha[MeOH]} \tag{57}$$

$$[H_2O] = \frac{\alpha[MeOH][DMC]}{[MB] + \alpha[MeOH]} \tag{58}$$

A system of two ODEs Equations (35) and (39) can be solved by employing Equations (55)–(58). Thus, the problem can in principle be solved as an ODE problem coupled to algebraic equations.

### 2.3.3. Transformation to ODEs

In order to obtain a more robust algorithm for parameter estimation, the differential-algebraic problem is transformed to ODEs as follows.

From Equation (48),

$$\frac{[BG]}{[DMC]} = \frac{\frac{MB}{MeOH}}{\frac{MB}{MeOH} + \alpha} \tag{59}$$

$$[BG] = \frac{y}{\alpha + y}[DMC] = f(y)[DMC] \tag{60}$$

where $y = \frac{[MB]}{[MeOH]}$.

The procedure is continued by differentiation as shown below,

$$\frac{df}{dt} = \frac{\frac{dy}{dt}(\alpha + y) - \frac{dy}{dt}(y)}{(\alpha + y)^2} = \frac{\alpha \frac{dy}{dt}}{(\alpha + y)^2} \tag{61}$$

$$\frac{d[BG]}{dt} = \frac{df}{dt}[DMC] + f\frac{d[DMC]}{dt} \tag{62}$$

$$\frac{d[BG]}{dt} = \frac{\alpha[DMC]}{(\alpha + y)^2}\frac{dy}{dt} + \frac{y}{\alpha + y}\frac{d[DMC]}{dt} \tag{63}$$

The definition $y$ gives the relation

$$\frac{d[MB]}{dt} = \frac{dy}{dt}[MeOH] + y\frac{d[MeOH]}{dt} \tag{64}$$

$$\frac{dy}{dt} = \frac{\frac{d[MB]}{dt}}{[MeOH]} - y\frac{\frac{d[MeOH]}{dt}}{[MeOH]} \tag{65}$$

Equation (16) gives

$$\frac{d[MeOH]}{dt} = -\frac{d[MB]}{dt} - 2r_2\rho_B \tag{66}$$

A combination of Equations (65) and (66) gives

$$\frac{dy}{dt} = \frac{\frac{d[MB]}{dt}}{[MeOH]} + y\frac{\left(\frac{d[MB]}{dt} + 2r_2\rho_B\right)}{[MeOH]} \tag{67}$$

Equations (41) and (63) give

$$\begin{aligned}\frac{d[BG]}{dt} &= r_4 - \frac{d[MB]}{dt}\\ &= \frac{\alpha[DMC]}{(\alpha+y)^2}\frac{dy}{dt} + \frac{y}{\alpha+y}r_2\rho_B\end{aligned} \tag{68}$$

From Equation (68) $dy/dt$ is solved:

$$\begin{aligned}\frac{dy}{dt} &= \frac{(\alpha+y)^2}{\alpha[DMC]}\\ &\left(r_4 - \frac{y}{\alpha+y}r_2\rho_B - \frac{d[MB]}{dt}\right)\end{aligned} \tag{69}$$

Equations (67) and (69) are equal, therefore,

$$\begin{aligned}&\frac{(1+y)}{[MeOH]}\frac{d[MB]}{dt} + 2r_2\rho_B\frac{y}{[MeOH]}\\ &= \frac{(\alpha+y)^2}{\alpha[DMC]}\left(r_4 - \frac{y}{\alpha+y}r_2\rho_B\right) - \frac{(\alpha+y)^2}{\alpha[DMC]}\frac{d[MB]}{dt}\end{aligned} \tag{70}$$

from which the time derivative is solved explicitly:

$$\begin{aligned}&\left[\frac{(1+y)[DMC]}{[MeOH]} + \frac{(\alpha+y)^2}{\alpha}\right]\frac{d[MB]}{dt}\\ &= -\left[\frac{(\alpha+y)y}{\alpha} - \frac{2y[DMC]}{[MeOH]}\right]r_2\rho_B + \frac{(\alpha+y)^2}{\alpha}r_4\end{aligned} \tag{71}$$

$$\begin{aligned}&\left[\frac{\alpha(1+y)[DMC]}{(\alpha+y)^2[MeOH]} + 1\right]\frac{d[MB]}{dt}\\ &= r_4 - \frac{y}{\alpha+y}\left[1 - \frac{2\alpha}{\alpha+y}\frac{[DMC]}{[MeOH]}\right]r_2\rho_B\end{aligned} \tag{72}$$

The final forms of the mass balances for the liquid-phase components in the batch reactor thus become ($\alpha$ = constant):

$$\frac{d[DMC]}{dt} = r_2 \rho_B \tag{73}$$

$$\frac{d[BO]}{dt} = -r_4 \tag{74}$$

$$\frac{d[MB]}{dt} = \omega^{-1} \left\{ r_4 - \left[ \frac{y}{\alpha+y} \right] \left[ 1 - \frac{2\alpha}{\alpha+y} \frac{[DMC]}{[MeOH]} \right] r_2 \rho_B \right\}$$
$$\omega = \left[ \frac{\alpha(1+y)[DMC]}{(\alpha+y)^2[MeOH]} + 1 \right] y = \frac{[MB]}{[MeOH]} \tag{75}$$

$$\frac{d[BG]}{dt} = r_4 - \frac{d[MB]}{dt} \tag{76}$$

$$\frac{d[H_2O]}{dt} = r_2 \rho_B - r_4 + \frac{d[MB]}{dt} \tag{77}$$

$$\frac{d[MeOH]}{dt} = -2r_2 \rho_B - \frac{d[MB]}{dt} \tag{78}$$

The rates of the rate-determining steps ($r_2$ and $r_4$) were obtained from the mechanism—these rates include only the concentrations of $CO_2$, MeOH, DMC, MB, BO, BG and $H_2O$, respectively (Appendix A: Derivation of the Rate Equations). The rate equations are given below. The ODEs were solved with the backward difference method during the parameter estimation, which was performed with the Levenberg–Marquardt algorithm [7].

Rate equations:

$$r_2 = \frac{k_2 \left( [MeOH]P_{CO_2} - \frac{1}{K} \frac{[DMC][BG]}{[MB]} \right)}{1 + K_1[MeOH] + \frac{1}{K_3} \frac{[DMC][BG]}{[MB]}} \tag{79}$$

$$r_4 = k_4[BO][MeOH] \tag{80}$$

## 2.3.4. Parameter Estimation Results

Rate parameters $k_2$ and $k_4$ were estimated from isothermal experiments shown as Equations (79) and (80). The experimental temperature was 150 °C, and the pressure was 45 bar of $CO_2$ initially. Parameters $K_1$ and $1/K_3$ (Table 3, Equations (47) and (48)) turned out not to be significant and were approximated to zero in the rate expression $r_2$. Parameter $\alpha$ was determined separately from the plot according to Equation (47); $\alpha = 1.3 \times 10^{-3}$. The thermodynamic equilibrium constant was estimated from theoretical calculations: $K = 0.08 \times 10^{-5}$, at 150 °C.

**Table 3.** Parameter estimation results.

| Parameter | Value | Error/% |
|:---:|:---:|:---:|
| $k_2$ | $1.60 \times 10^{-5}$ | 6.8 |
| $k_4$ | $1.12 \times 10^{-2}$ | 6.7 |
| $K_1 = 0$, $1/K_3 = 0$, $\alpha = 1.3 \times 10^{-3}$, $K = 0.08 \times 10^{-5}$, at 150 °C | | |

The solution of ODEs and parameter estimation progressed very smoothly and led to an excellent description of the experimental data. The errors of the kinetic parameters were clearly less than 10%. The performance of the procedure is illustrated in Figure 11 The numerical values of the estimated parameters are enlisted in Table 3. The overall degree of explanation was 99.98%.

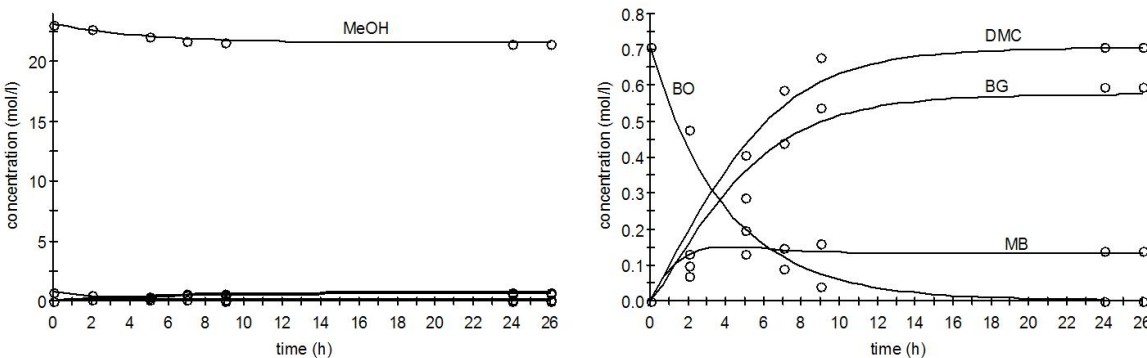

**Figure 11.** Performance of the solution of ordinary differential equations (ODEs) and parameter estimation method: synthesis of dimethyl carbonate (DMC) (150 °C, initial $CO_2$ pressure of 45 bar). The smaller concentrations in the left figure (butylene oxide (BO), DMC, butylene glycol (BG), methylene butylate (MB)) are magnified in the right figure.

## 3. Conclusions

Solution of a differential-algebraic problem in connection to both fast and slow reaction steps is a demanding task when coupled to a parameter estimation task. In order to surmount the numerical problems often appearing for DAE systems, we propose a robust procedure, which implies the transformation of the DAE system to a set of explicit ODEs that can be easily solved in-situ during the estimation of kinetic parameters. The success of the methodology was illustrated with two generic examples and a case study, synthesis of dimethyl carbonate (DMC) from methanol and carbon dioxide.

**Author Contributions:** J.-P.M. and V.E. conceived and designed the experiments; V.E. performed the experiments; T.S., J.W., D.M., J.-P.M. and V.E. analyzed the data; J.W. and E.T. performed the numerical simulations; T.S. and E.T. wrote the paper. All authors have read and agreed to the published version of the manuscript.

**Funding:** This research was funded by Academy of Finland (Academy professor's grant number 319002).

**Conflicts of Interest:** The authors declare no conflict of interest.

## Notation

| | |
|---|---|
| c | concentration |
| $c^*$ | concentration of an intermediate |
| $f$ | function |
| $k$ | reaction rate constant |
| $r$ | Rate |
| $t$ | Time |
| $y$ | concentration variable |
| $\alpha$ | parameter in rate equation |
| $\beta$ | parameter in rate equation |
| $\gamma$ | merged concentration |
| $\rho_B$ | catalyst bulk density (mass of catalyst-to-liquid volume) |
| $\omega$ | merged parameter |
| [] | concentration |

## Appendix A. Derivation of the Rate Equations

$$MeOH + * = MeOH* \tag{A1}$$

This rapid step is in quasi-equilibrium giving

$$K_1 = \frac{[MeOH*]}{[MeOH][*]} \tag{A2}$$

from which the surface concentration is solved,

$$[MeOH*] = K_1[MeOH] \tag{A3}$$

Analogously, for the rapid step

$$MB + MC* = DMC + BG + * \tag{A4}$$

can be written

$$K_3 = \frac{[DMC][BG][*]}{[MC*][MB]} \tag{A5}$$

and the surface concentration of MC is solved,

$$[MC*] = \frac{[DMC][BG][*]}{K_3[MB]} \tag{A6}$$

The total balance of sites on the catalyst surface is ($0*$ is the total concentration of available siters on the surface)

$$[MeOH*] + [MC*] + [*] = [0*] \tag{A7}$$

Equations (A3), (A6) and (A7) give the concentration of vacant sites,

$$[*] = \frac{[0*]}{K_1[MeOH] + [DMC][BG]/(K_3[MB])} \tag{A8}$$

For the slow, rate determining step 2 is valid:

$$r_2 = k_2'[MeOH*]P_{CO2} - k_{-2}'[MC*] \tag{A9}$$

$$r_2 = (k_2'K_1[MeOH]P_{CO2} - k_{-2}'\frac{[DMC][BG]}{K_3[MB]})[*] \tag{A10}$$

The following merged constants are introduced

$$k_2'/k_{-2}' = K_2,$$
$$K_1K_2K_3 = K,$$
$$k_2'K_1[0*] = k_2$$

and introduced in Equation (A10). The final rate expression for step 2 becomes

$$r_2 = \frac{k_2([MeOH]P_{CO2} - \frac{[DMC][BG]}{K[MB]})}{1 + K_1[MeOH] + \frac{[DMC][BG]}{K_3[MB]}} \tag{A11}$$

The slow step

$$BO + MeOH = MB \tag{A12}$$

takes place in the liquid bulk (not on the catalyst surface), and the rate is expressed in a straightforward way,

$$r_4 = k_4[BO][MeOH] \tag{A13}$$

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
