# Peer review of "A Robust Method for the Estimation of Kinetic Parameters for Systems Including Slow and Rapid Reactions—From Differential-Algebraic Model to Differential Model"

_processes, doi:10.3390/pr8121552_

Round 1

Reviewer 1 Report

In the manuscript entitled "A ROBUST METHOD FOR THE ESTIMATION OF KINETIC PARAMETERS FOR SYSTEMS INCLUDING SLOW AND RAPID REACTIONS" the authors have developed a method for estimating the kinetics parameters that allows to obtain concentration profiles of reaction mixtures  that easily shows how the reaction mixture evolves in time. This is an important topic that would be interesting for a wide audience of experimental chemists. However I have some questions  about the model that, in my opinion,  should be addressed prior publication. 

1) The authors discriminate between slow and rapid individual reaction steps, but they have not considered the reversibility for irreversibility nature of these steps. For instance, in Example 2.1, the arrows used in the chemical scheme (named equilibrium arrows) imply that the two steps are reversible. It would imply that 

d[A]/dt =-r(1)+r(-1) 

where r(1) corresponds to the rate constant associated with the A->R forward transformation, whereas r(-1) /not included/ would correspond to the rate constant associated with R->A reverse process. For the intermediate R, the system would be more complicated:

d[R]/dt= r(1)+r(-2)-r(2)-r(-1)

In fact, in the two initial  (2.1 and 2.2) scenarios considered in the manuscript it is assumed irreversible single steps. It should be indicated in the description of the systems. I suggest, in order to make the model more realistic, assume  that  reactions can be also reversible. In any case, appropriate arrows should be used in the model reaction schemes.

2) Synthesis of dimethyl carbonate:

-I asume that * represent the catalyst? please specify

-d[DMC]/dt=(r3+r5) rhoB

If rhoB is related to the catalyst bulk density, I assume is something related to the 'catalyst concentration'. In that case, r3 should include the concentration of MC*, and therefore only affected by the catalyst once, but r5 is related to MC* and MeOH*, and it should be related to rhoB^2, isn't  it?. Maybe if the authors include all the mathematical steps in some kind of Supporting Information document these doubts would be easily solved because similar doubts can arise from the other species. In fact, in the proposed scheme, only step 5 seems to be irreversible, as commented in the text, therefore, the reversibility of all other steps should be included in the mathematical equations.

3) Please use different symbols or colors in Figure 8 to help the reader identifícate the original set of points that corresponds to each curve.

Author Response

Reviewer I

In the manuscript entitled "A ROBUST METHOD FOR THE ESTIMATION OF KINETIC PARAMETERS FOR SYSTEMS INCLUDING SLOW AND RAPID REACTIONS" the authors have developed a method for estimating the kinetics parameters that allows to obtain concentration profiles of reaction mixtures  that easily shows how the reaction mixture evolves in time. This is an important topic that would be interesting for a wide audience of experimental chemists. However I have some questions  about the model that, in my opinion,  should be addressed prior publication. 

1) The authors discriminate between slow and rapid individual reaction steps, but they have not considered the reversibility for irreversibility nature of these steps. For instance, in Example 2.1, the arrows used in the chemical scheme (named equilibrium arrows) imply that the two steps are reversible. It would imply that 

d[A]/dt =-r(1)+r(-1) 

where r(1) corresponds to the rate constant associated with the A->R forward transformation, whereas r(-1) /not included/ would correspond to the rate constant associated with R->A reverse process. For the intermediate R, the system would be more complicated:

d[R]/dt= r(1)+r(-2)-r(2)-r(-1)

In fact, in the two initial  (2.1 and 2.2) scenarios considered in the manuscript it is assumed irreversible single steps. It should be indicated in the description of the systems. I suggest, in order to make the model more realistic, assume  that  reactions can be also reversible. In any case, appropriate arrows should be used in the model reaction schemes.

*** We use de facto the reversible kinetics, because both the forward and backward reactions are incorporated in the rate expressions (4), (5), (15) and (16). Our r1=r1(+) - r1(-). For example, equation (1) can alternatively be written as r1=k(+1)*cA-k(-1)*cR, but for elementary reactions the forward and backward rate constants are related by the equilibrium constant of the step, i.e. K=k(+)/k(-), which means that k(-)=k(+)/K, therefore r1 can be expressed as r1=k(+1)*(cA-cR/K1) in equation (4). The same is valid for equations (4), (5), (15) and (16).

2) Synthesis of dimethyl carbonate:

-I assume that * represent the catalyst? please specify

***YES, * denotes an active site on the catalyst surface. It is explained in the revision now.

-d[DMC]/dt=(r3+r5) rhoB

If rhoB is related to the catalyst bulk density, I assume is something related to the 'catalyst concentration'. In that case, r3 should include the concentration of MC*, and therefore only affected by the catalyst once, but r5 is related to MC* and MeOH*, and it should be related to rhoB^2, isn't  it?. Maybe if the authors include all the mathematical steps in some kind of Supporting Information document these doubts would be easily solved because similar doubts can arise from the other species. In fact, in the proposed scheme, only step 5 seems to be irreversible, as commented in the text, therefore, the reversibility of all other steps should be included in the mathematical equations.

*** rhoB is simply the mass of catalyst –to- liquid mass in the reactor. We have included the details of the derivation of the rate equations (79) and (80) in an Appendix (supporting information), so that the chemical and mathematical background of the kinetic model becomes clear.

3) Please use different symbols or colors in Figure 8 to help the reader identifícate the original set of points that corresponds to each curve.

*** Figure 8 has been checked and clarified. Extra figures which repeat the same information has been removed from the figure.

Reviewer 2 Report

The present manuscript is well written and clear as to the information displayed. However I cannot recommend its publication in Processes. The manuscript does not stop representing a good kinetic problem, suitable as an exercise for students in the last year of the Chemistry degree or for a Master with contents of chemical kinetics. The problem posed is not new and there are currently numerous software solutions on the market (both commercial and free) that solve the questions raised in the manuscript without any problem of lack of robustness in the solutions to the differential equations that arise in this class. of mechanisms.

I would recommend that the authors send this manuscript to a journal dedicated to chemistry education such as J. Chem. Edu.

Author Response

Reviewer II

The present manuscript is well written and clear as to the information displayed. However I cannot recommend its publication in Processes. The manuscript does not stop representing a good kinetic problem, suitable as an exercise for students in the last year of the Chemistry degree or for a Master with contents of chemical kinetics. The problem posed is not new and there are currently numerous software solutions on the market (both commercial and free) that solve the questions raised in the manuscript without any problem of lack of robustness in the solutions to the differential equations that arise in this class. of mechanisms.

I would recommend that the authors send this manuscript to a journal dedicated to chemistry education such as J. Chem. Edu.

*** We agree that the problem is not new,and in principle the numerical problem could be solved by software designed for differential-algebraic systems, but in case of parameter estimation the issue is not straightforward, because the the rate parameters are unknown in the beginning of the optimization process and when the DAE solver happens to arrive in a domain of unrealistic values of parameters, the algebraic equations might not any more have solutions for realistic concentration values and the entire algorithm collapses. Therefore, the solution of the entire problem as a system of ODEs is more safe. We have selected simple examples (consecutive and parallel) reactions to illustrate the basic principle, and then solved a complex research case, the formation kinetics of dimethyl carbonate. The manuscript has been written for the research audience, not for basic education purposes. Therefore we think that ‘Processes’ is the appropriate publication forum for the work.

Reviewer 3 Report

The work is devoted to the problem of determining the kinetic parameters of reaction systems in which both very fast and very slow reactions occur. Then there are mathematical problems of a differential-algebraic nature. The proposed method of transforming a differential-algebraic problem into a differential one is well known and is another form of the method that uses algebraic relationships to explicitly determine and then eliminate other variables. As far as the essence of the issue is concerned, the work does not bring any news.

The title of the paper suggests the estimation of kinetic parameters, but this issue is devoted to only one comment concerning the application of the Levenberg-Marquardt algorithm. In fact, this article deals with  transformation of  an differential-algebraic problem into a differential one.

The analysis conducted for two simple cases is very basic and is often discussed in student chemical kinetics courses.

The most interesting part of the work concerns the synthesis of dimethyl carbonate and a broad analysis of this problem could significantly increase the value of the work. The assumptions made in formulating the kinetic model should be deeply discussed. The reaction system is a heterogeneous system and the assumptions made are not obvious.

In the last members of equations 10 and 11, the sign "-" should be changed to the sign "+". This error does not occur in further equations.

Author Response

Reviewer III

The work is devoted to the problem of determining the kinetic parameters of reaction systems in which both very fast and very slow reactions occur. Then there are mathematical problems of a differential-algebraic nature. The proposed method of transforming a differential-algebraic problem into a differential one is well known and is another form of the method that uses algebraic relationships to explicitly determine and then eliminate other variables. As far as the essence of the issue is concerned, the work does not bring any news.

The title of the paper suggests the estimation of kinetic parameters, but this issue is devoted to only one comment concerning the application of the Levenberg-Marquardt algorithm. In fact, this article deals with  transformation of  an differential-algebraic problem into a differential one.

*** We agree that the main point is the transformation of the DAE model to an ODE model, but this transformation is particularly useful when dealing with parameter estimation (see our reply to Reviewer II above). The title has been modified as indicated by the Reviewer.

The analysis conducted for two simple cases is very basic and is often discussed in student chemical kinetics courses.

*** We agree that the consecutive and parallel reactions are treated in basic textbooks and also the stationary state hypothesis are described in textbook of chemical kinetics and chemical reaction engineering, but our approach has the emphasis on the transformation of DAE models to ODE models, but still keeping the stationarity principle (quasi-steady state) valid.

The most interesting part of the work concerns the synthesis of dimethyl carbonate and a broad analysis of this problem could significantly increase the value of the work. The assumptions made in formulating the kinetic model should be deeply discussed. The reaction system is a heterogeneous system and the assumptions made are not obvious.

*** We have included an explanation of the detailed reaction mechanism and the derivation of the rate equations (79)-(80) in an Appendix. A similar proposal was given by Reviewer I.

In the last members of equations 10 and 11, the sign "-" should be changed to the sign "+". This error does not occur in further equations.

*** Thank you, we have noticed and corrected the misprint.

Reviewer 4 Report

My recommendation is to publis this paper with minor revision

My main concern is the following one:

In the paper, it must be discussed the hidden computational drawback related to the accuracy of the described robust method. The proposed approach is based on the replacing the mixed system of ordinaty differential equations and nonlinear algebraic  equations by the system of ordinary differential equations. (The system of algebraic equations is based on the quasi-steady-state hypothesis). Then, the new system of differential equations is integrated. However, the computational error is accumulated during the interation, ican be dangerous for the validity of the assumed quas-steady-state hypothesis.

Consequently, I propose to discuss it in the revised version of the paper.

Secondly. I attach some our papers especially 

  • Branco Pinto, G.Yablonsky, G.B. Marin, and D. Constales, “The Switching

Point between the Kinetic and Thermodynamic Control”, Comp.Chem. Eng., 2017

In this paper, authors studied two consecutive reactions A=B=C . I appreciate referring this paper if it will be needed.  

Author Response

Reviewer IV

My recommendation is to publish this paper with minor revision

My main concern is the following one:

In the paper, it must be discussed the hidden computational drawback related to the accuracy of the described robust method. The proposed approach is based on the replacing the mixed system of ordinary differential equations and nonlinear algebraic  equations by the system of ordinary differential equations. (The system of algebraic equations is based on the quasi-steady-state hypothesis). Then, the new system of differential equations is integrated. However, the computational error is accumulated during the integration, it can be dangerous for the validity of the assumed quasi-steady-state hypothesis.

*** The integration of the ODEs created is very precise, because an excellent ODE solution code is used, backward difference method designed for stiff ODEs. In fact we do not violate against the stationarity (quasi-steady state) hypothesis, because the differentiations which are carried out in the modelling are formal, they are always valid for the particular system. For example, by comparing figures 1 and 2, it can be seen that high values are needed for the rate constant k2 to fully justify the use of pseudo-state hypotheis (k2= 100 or higher). The concentrations displayed in figure 1 have exactly the same values if they are solved from the DAE model as they obtain from our ODE model.

Consequently, I propose to discuss it in the revised version of the paper.

Secondly. I attach some our papers especially 

  • Branco Pinto, G.Yablonsky, G.B. Marin, and D. Constales, “The Switching

Point between the Kinetic and Thermodynamic Control”, Comp.Chem. Eng., 2017

In this paper, authors studied two consecutive reactions A=B=C . I appreciate referring this paper if it will be needed.  

*** We have checked this article and included it in the reference list. We have also added a comment on the kinetic and thermodynamic control.

Round 2

Reviewer 1 Report

The last version of the manuscript has been improved.

I  acknowledge the authors for clarifying all the comments/issues addressed. The results are sound, and after a careful reading, I have no more comments to add.

Reviewer 2 Report

Sorry, bit I hace not new coments

Reviewer 3 Report

I accept the authors' explanations.